# MARICH: A QUERY-EFFICIENT MAX-INFORMATION MODEL EXTRACTION ATTACK USING PUBLIC DATA

## ABSTRACT

In this paper, we study black-box model stealing attacks where the attacker is only able to query a machine learning model only through publicly available APIs. Specifically our aim is to design a black-box model stealing attack that uses a minimal number of queries to create an informative replica of the target model. First, we reduce this problem into an online variational optimisation problem. The attacker solves this problem to select the most informative queries that maximise the entropy of the selected queries and simultaneously reduce the mismatch between the target and the stolen models. We propose an online and adaptive algorithm, MARICH that leverages active learning to select the queries. We instantiate efficiency of our attack on different text and image data sets and different models including BERT and ResNet18. Marich is able to steal a model that can achieve 69-96% of true model's accuracy using 1,070 - 6,950 samples from the attack datasets which are completely different from the training data sets. Our stolen models also achieve 85-95% accuracy of membership inference and also show 77-94% agreement of membership inference with direct membership inference on the target models. Our experiments validate that Marich is query efficient and also capable of creating an informative replica of the target model.

## 1 INTRODUCTION

In recent years, Machine Learning as a Service (MLaaS) are widely deployed and used in industries. In MLaaS (Ribeiro et al., 2015), an ML model is trained remotely on a private dataset, deployed in a Cloud, and offered for public access through a prediction API, such as Amazon AWS, Google API, Microsoft Azure. This API allows an user, including a potential adversary, to send queries to the ML model and fetch corresponding predictions. Recent works have shown such models with public APIs can be stolen or extracted by designing black-box model extraction attacks (Tramèr et al., 2016). In model extraction attacks, an adversary queries the target model with a query dataset, which might be same or different than the private dataset, collects the corresponding predictions from the target model, and builds a replica model of the target model. The goal is to construct a model which is nearly-equivalent to the target model over the input space of interest (Jagielski et al., 2020).

Often, ML models are proprietary, guarded by IP rights, and expensive to build. These models might be trained on datasets which are expensive to obtain (Yang et al., 2019) and consist of private data of individuals (Lowd & Meek, 2005). Also, extracted models can be used to perform other privacy attacks on the private dataset used for training, such as membership inference (Nasr et al., 2019). Thus, understanding susceptibility of models accessible through MLaaS presents an important conundrum. This motivates us to investigate black-box model extraction attacks while the adversary has no access to the private data or a perturbed version of it (Papernot et al., 2017). Instead, the adversary uses a public dataset to query the target model (Orekondy et al., 2019; Pal et al., 2020).

Black-box model extraction attacks pose a tension between the number of queries sent to the target ML model and the accuracy of extracted model (Pal et al., 2020). With more number of queries and predictions, adversary can build a better replica. But querying an API too much can be expensive, as each query incurs a monetary cost in MLaaS. Also, researchers have developed algorithms that can detect adversarial queries, when they are not well-crafted or sent to the API in large numbers (Juuti et al., 2019; Pal et al., 2021). Thus, designing a query-efficient attack is paramount for practical deployment. Also, it indicates how more information can be leaked from a target model with less number of interactions.

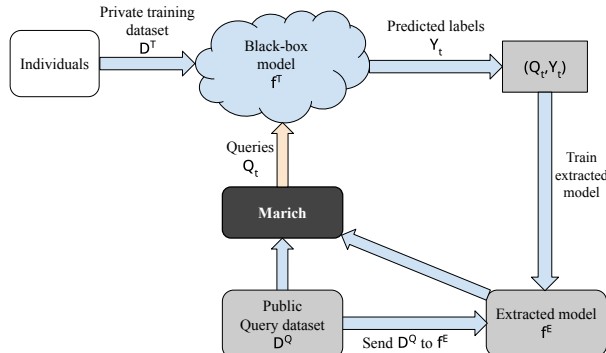

Figure 1: A schematic for black-box model extraction attack with MARICH.

In this paper, *we investigate effective definitions of efficiency of model extraction and corresponding algorithm design for query-efficient black-box model extraction attack with public data, which is oblivious to deployed model and applicable for any datatype.*

**Our contributions.** Our investigation yields three-fold contributions.

*1. Formalism: Distribution Equivalence and Max-Information Extraction.* Often, the ML models, specifically classifiers, are stochastic algorithms. They also include different elements of randomness during training. Thus, rather than focusing on equivalence of extracted and target models in terms of a fixed dataset or accuracy on that dataset (Jagielski et al., 2020), we propose a distributional notion of equivalence. We propose that if the joint distributions induced by a query generating distributions and corresponding prediction distributions due to the target and extracted models are same, they will be called distributionally equivalent Sec. 4). Another proposal is to reinforce the objective of the attack, i.e. to extract as much information as possible from the target model. This allows us to formulate the Max-Information attack, where the adversary aims to maximise the mutual information between the extracted and target models' distributions. We further show that both of these attacks can be performed by optimising a variational objective (Staines & Barber, 2012).

*2. Algorithm: Adaptive Query Selection for Extraction with* MARICH. We propose an algorithm, MARICH (Sec. 5), that optimises the objective of the variational optimisation problem (Equation 6). Given an extracted model, a target model, and previous queries, MARICH adaptively selects a batch of queries enforcing this objective. Then, it sends the queries to the target model, collect the predictions, and use them to further train the extracted model (Algorithm 1). In order to select the most informative set of queries, it deploys three sampling strategies sequentially. These three strategies select: a) the most informative set of queries, b) the most diverse set of queries in the first selection, and c) the set of queries in the first selection where the target and extracted models mismatch the most. Together these strategies allow MARICH to select a small subset of queries, which maximise the information leakage and align the extracted model with the target (Figure 1).

*3. Experimental Analysis.* We perform extensive experimental evaluation with both image and text datasets, and diverse model classes, such as logistic regression, ResNet18, and BERT (Sec. 6). Our experimental results validate that MARICH can extract accurate and informative replicas of the target models in comparison to random sampling. While MARICH uses a small number of queries $(0.83 - 6.15\%)$ selected from publicly available query datasets, the models extracted by it lead to comparable accuracy with the target model while encountering a membership inference attack. This shows that MARICH can extract alarmingly informative models query-efficiently.

## 2 RELATED WORKS

Here, we elaborate the questions in the model extraction literature that we aim to mitigate.

**Taxonomy of Model Extraction.** Black-box model extraction (or model stealing or model inference) attacks aim to *replicate* of a target ML model, commonly classifiers, deployed in a remote service and accessible through a public API (Tramèr et al., 2016). The replication is done in such a way that the extracted model achieves one of the three goals: a) accuracy close to that of the target model on the private training data used to train the target model, b) maximal agreement in predic-

tions with the target model on the private training data, and c) maximal agreement in prediction with the target model over the whole input domain. Depending on the objective, they are called *task accuracy*, *fidelity*, and *functional equivalence model extractions*, respectively (Jagielski et al., 2020). Here, *we generalise these three approaches using a novel definition of distributional equivalence and also introduce a novel information-theoretic objective of model extraction which maximises the mutual information between the target and the extracted model over the whole data domain.*

**Framework of Attack Design.** Following Tramèr et al. (2016), researchers have proposed multiple attack algorithms to perform one of the three types of model extraction. The attack algorithms are based on two main approaches: direct recovery (target model specific) (Milli et al., 2019; Batina et al., 2018; Jagielski et al., 2020) and learning (target model specific/oblivious). The learning-based approaches can also be categorised into supervised learning strategies, where the adversary has access to both the true labels of queries and the labels predicted by the target model (Tramèr et al., 2016; Jagielski et al., 2020), and online active learning strategies, where the adversary has only access to the predicted labels of the target model, and actively select the future queries depending on the previous queries and predicted labels (Papernot et al., 2017; Pal et al., 2020; Chandrasekaran et al., 2020). *As query-efficiency is paramount for an adversary while attacking an API to save the budget and to keep the attack hidden and also the assumption of access true label from the private data is restrictive, we focus on designing an online and active learning-based attack strategy which can be model oblivious.*

**Classes of Target Model.** While (Milli et al., 2019; Chandrasekaran et al., 2020) focus on performing attacks against linear models, all others are specific to neural networks (Milli et al., 2019; Jagielski et al., 2020; Pal et al., 2020) and even a specific architecture (Correia-Silva et al., 2018). In contrast, MARICH *is based on active learning methodology and also capable to attack both linear models and neural networks.*

**Types of Query Feedback.** Learning-based attack algorithms often assume access to either the probability vector of the target model over all the predicted labels (Tramèr et al., 2016; Orekondy et al., 2019; Pal et al., 2020; Jagielski et al., 2020), or the gradient of the last layer of the target neural network (Milli et al., 2019; Miura et al., 2021), which are hardly available in a public API. In contrast, following (Papernot et al., 2017), *we assume access to only the predicted labels of the target model for a given set of queries, which is always available through a public API.*

**Type of Query Dataset.** The adversary needs a query dataset to select the queries from and to send it to the target model to obtain predicted labels. In literature, researchers assume three types of query datsets: synthetically generated samples (Tramèr et al., 2016), adversarially perturbed private (or problem domain) dataset (Papernot et al., 2017; Juuti et al., 2019), and publicly available (or out-of-problem domain) dataset (Orekondy et al., 2019; Pal et al., 2020). As we do not want to restrict MARICH to have access to the knowledge of the private dataset or any perturbed version of it, *we use publicly available datasets, which are different than the private dataset.*

In brief, we propose an online and active-learning based model extraction attack using MARICH, which is model-oblivious, assumes only access to the predicted label for a given query through a public API, and uses only publicly available non-domain data to query the target model. This is a less restrictive setup than the ones considered in literature, while the models extracted by MARICH demonstrate significant accuracy and act as informative replica of the target model to conduct pervasive membership inference on the private dataset.

## 3 BACKGROUND: CLASSIFIERS, MODEL EXTRACTION, AND MEMBERSHIP INFERENCE ATTACKS

Before proceeding to the details of our contribution, we present the fundamentals of a classifier in ML, and two types of inference attacks: Model Extraction (ME) and Membership Inference (MI).

**Classifiers.** A classifier in ML (Goodfellow et al., 2016) is a function $f : \mathcal{X} \to \mathcal{Y}$ that maps a set of input features $\mathbf{X} \in \mathcal{X}$ to an output $Y \in \mathcal{Y}$.[1] The output space is a finite set of classes, i.e. $\{1, \ldots, k\}$. Specifically, a classifier $f$ is a parametric function, denoted as $f_\theta$, with parameters $\theta \in \mathbb{R}^d$, and is trained on a dataset $\mathbf{D}^T$, i.e. a collection of $n$ tuples $\{(\mathbf{x}_i, y_i)\}_{i=1}^n$ generated IID

---

[1]We represent sets/vectors by bold letters, and the corresponding distributions by calligraphic letters. We express random variables in uppercase, and an assignment of a random variable in lowercase.

from an underlying distribution $\mathcal{D}$. Training implies that given a model class $\mathcal{F} = \{f_\theta | \theta \in \Theta\}$, a loss function $l : \mathcal{Y} \times \mathcal{Y} \to \mathbb{R}_{\geq 0}$, and training dataset $\mathbf{D}^T$, we aim to find the optimal parameter $\theta^* \triangleq \arg\min_{\theta \in \Theta} \sum_{i=1}^n l(f_\theta(\mathbf{x}_i), y_i)$. We use cross-entropy, i.e. $l(f_\theta(\mathbf{x}_i), y_i) \triangleq -y_i \log(f_\theta(\mathbf{x}_i))$, as the loss function for classification.

**Model Extraction Attack.** A model extraction attack is an inference attack where an adversary aims to steal a target model $f^T$ trained on a private dataset $\mathbf{D}^T$ and create another replica of it $f^E$ (Tramèr et al., 2016). In the black-box setting that we are interested in, the adversary can only query the target model $f^T$ by sending queries $Q$ through a publicly available API and to use the corresponding predictions $\hat{Y}$ to construct $f^E$. The goal of the adversary is to create a model which is either (a) as similar to the target model as possible for all input features, i.e. $f^T(x) = f^E(x) \ \forall x \in \mathcal{X}$ (Song & Shmatikov, 2020; Chandrasekaran et al., 2020) or (b) predicts labels that has maximal agreement with that of the labels predicted by the target model for a given data-generating distribution, i.e. $f^E = \arg\min \Pr_{x \sim \mathcal{D}}[l(f^E(x), f^T(x))]$ (Tramèr et al., 2016; Pal et al., 2020; Jagielski et al., 2020). The first type of attacks are called the functionally equivalent attacks. The later family of attacks is referred as the fidelity extraction attacks. The third type of attacks aim to find an extracted model $f^E$ that achieves maximal classification accuracy for the underlying private dataset used to train the $f^T$. These are called task accuracy extraction attacks (Tramèr et al., 2016; Milli et al., 2019; Orekondy et al., 2019). In this paper, *we generalise the first two type of attacks by proposing the distributionally equivalent attacks and experimentally show that it achieves significant task accuracy.*

**Membership Inference Attack.** Another popular family of inference attacks on ML models is the Membership Inference (MI) attacks (Shokri et al., 2017; Yeom et al., 2018). In MI attacks, given a private (or member) dataset $\mathbf{D}^T$ to train $f^T$ and another non-member dataset $S$ with $|\mathbf{D}^T \cap S| \neq \emptyset$, the goal of the adversary is to infer whether any $x \in \mathcal{X}$ is sampled from the member dataset $\mathbf{D}^T$ or the non-member dataset $S$. Effectiveness of an MI attacks can be measured by its accuracy of MI, i.e. the total fraction of times the MI adversary identifies the member and non-member data points correctly. Accuracy of MI attack on the private data using $f^E$ rather than $f^T$ is considered as a measure of effectiveness of the extraction attack (Nasr et al., 2019). We show that the model $f^E$ extracted using MARICH allows us to obtain similar MI accuracy as that obtained by directly attacking the target model $f^T$ using even larger number of queries. This validates that *the model $f^E$ by MARICH in a black-box setting acts as an information equivalent replica of the target model $f^T$.*

## 4 DISTRIBUTIONAL EQUIVALENCE & MAX-INFORMATION MODEL EXTRACTIONS: A VARIATIONAL OPTIMISATION FORMULATION

In this section, we introduce the notions of distributionally equivalent and max-information model extractions. We further reduce both the attacks into a single variational optimisation problem.

**Definition 1** (Distributionally Equivalent Model Extraction). *For any query generating distribution $\mathcal{D}^Q$ over $\mathbb{R}^d \times \mathcal{Y}$, an extracted model $f^E : \mathbb{R}^d \to Y$ is distributionally equivalent to a target model $f^T : \mathbb{R}^d \to Y$ if the joint distributions of input features $Q \in \mathbb{R}^d \sim \mathcal{D}^Q$ and predicted labels induced by both the models are same almost surely. This means that for any divergence $D$, two distributionally equivalent models $f^E$ and $f^T$ satisfy $D(\Pr(f^T(Q), Q) \| \Pr(f^E(Q), Q)) = 0 \ \forall \mathcal{D}^Q$.*

To ensure query-efficiency in distributionally equivalent model extraction, an adversary aims to choose a query generating distribution $\mathcal{D}^Q$ that minimises it further. If we assume that the extracted model is also a parametric function, i.e. $f^E_\omega$ with parameters $\omega \in \Omega$, we can solve the query-efficient distributionally equivalent extraction by computing

$$(\omega^*_{\text{DEq}}, \mathcal{D}^Q_{\min}) \triangleq \arg\min_{\omega \in \Omega} \arg\min_{\mathcal{D}^Q} D(\Pr(f^T_{\theta^*}(Q), Q) \| \Pr(f^E_\omega(Q), Q)). \tag{1}$$

Equation 1 allows us to choose a different class of models with different parametrisation for extraction till the joint distribution induced by it matches with that of the target model. For example, the extracted model can be a logistic regression or a CNN if the target model is a logistic regression. This formulation also enjoys the freedom to choose the data distribution $\mathcal{D}^Q$ for which we want to test the closeness. Rather the distributional equivalence pushes us to find the best query distribution for which the mismatch between the posteriors reduces the most and to compute an extracted model $f^E_{\omega^*}$ that induces the joint distribution closest to that of the target model $f^T_{\theta^*}$.

**Remark 1.** *If we choose $\mathcal{D}_{\min}^Q = \mathcal{D}^T$, distributional equivalence reduces to the fidelity extraction attack. If we choose $\mathcal{D}_{\min}^Q = \mathrm{Unif}(\mathcal{X})$, distributional equivalent extraction coincides with functional equivalent extraction. Thus, distributional equivalence attack can ensure both fidelity and functional equivalence extractions depending on the choice of query generating distribution $\mathcal{D}^Q$.*

**Theorem 1** (Upper Bounding Distributional Closeness). *If we choose KL-divergence as the divergence function $D$, then for a given query generating distribution $\mathcal{D}^Q$*

$$D_{\mathrm{KL}}(\Pr(f_{\theta^*}^T(Q), Q) \| \Pr(f_{\omega_{\mathrm{DEq}}^*}^E(Q), Q)) \leq \min_\omega \mathbb{E}_Q[l(f_{\theta^*}^T(Q), f_\omega^E(Q))] - H(f_\omega^E(Q)). \quad (2)$$

By variational principle, Theorem 1 implies that *minimising the upper bound* on the RHS will lead to an extracted model which minimises the KL-divergence for a chosen query distribution.

**Max-Information Model Extraction.** The common objective of any inference attack is to leak as much information as possible from the target model $f^T$. Specifically, in model extraction attacks, we want to create an informative replica $f^E$ of the target model $f^T$ such that it induces a joint distribution $\Pr(f_\omega^E(Q), Q)$ which retains the most information regarding the target's joint distribution. As adversary can control the query distribution, we want to choose such a query distribution $\mathcal{D}^Q$ for which the information leakage is maximised.

**Definition 2** (Max-Information Model Extraction). *A model $f^E : \mathbb{R}^d \to Y$ and query distribution $\mathcal{D}^Q$ are called a max-information extraction of a target model $f^T : \mathbb{R}^d \to Y$ and max-information query distribution, respectively, if they maximise the mutual information between the joint distributions of input features $Q \in \mathbb{R}^d \sim \mathcal{D}^Q$ and predicted labels induced by $f^E$ and that of the target model. Mathematically, $(f_{\omega^*}^E, \mathcal{D}_{\max}^Q)$ is a max-information extraction of $f_{\theta^*}^T$ such that*

$$(\omega_{\mathrm{MaxInf}}^*, \mathcal{D}_{\max}^Q) \triangleq \arg\max_\omega \arg\max_{\mathcal{D}_Q} I(\Pr(f_{\theta^*}^T(Q), Q) \| \Pr(f_\omega^E(Q), Q)) \quad (3)$$

Similar to Definition 1, Definition 2 also does not restrict us to choose a parametric model $\omega$ different than that of the target $\theta$. It also allows us to compute the data distribution $\mathcal{D}^Q$ for which the information leakage is maximum rather than relying on the private dataset $\mathbf{D}^T$ used for training $f^T$.

**Theorem 2** (Lower Bounding Information Leakage). *For any given distribution $\mathcal{D}^Q$, the information leaked by any max-information attack (Equation 3) is lower bounded as follows:*

$$I(\Pr(f_{\theta^*}^T(Q), Q) \| \Pr(f_{\omega_{\mathrm{MaxInf}}^*}^E(Q), Q)) \geq \max_\omega -\mathbb{E}_Q[l(f_{\theta^*}^T(Q), f_\omega^E(Q))] + H(f_\omega^E(Q)). \quad (4)$$

By variational principle, Theorem 2 implies that *maximising the lower bound* in the RHS will lead to an extracted model which maximises the mutual information between target and extracted joint distributions for a given query generating distribution.

**Distributionally Equivalent and Max-Information Extractions: A Variational Optimisation Formulation.** From Theorem 1 and 2, we observe that the lower and upper bounds of the objective functions of distribution equivalent and max-information attacks are negatives of each other. Specifically, $-D_{\mathrm{KL}}(\Pr(f_{\theta^*}^T(Q), Q) \| \Pr(f_{\omega_{\mathrm{DEq}}^*}^E(Q), Q)) \geq \max_\omega -F(\omega, \mathcal{D}^Q)$ and $I(\Pr(f_{\theta^*}^T(Q), Q) \| \Pr(f_{\omega_{\mathrm{MaxInf}}^*}^E(Q), Q)) \geq \max_\omega F(\omega, \mathcal{D}^Q)$, where

$$F(\omega, \mathcal{D}^Q) \triangleq -\mathbb{E}_Q[l(f_{\theta^*}^T(Q), f_\omega^E(Q))] + H(f_\omega^E(Q)). \quad (5)$$

Thus, following a variational approach, we aim to solve an optimisation problem on $F(\omega, \mathcal{D}^Q)$ in an online and frequentist manner. Specifically, we do not assume a parametric family of $\mathcal{D}^Q$. Instead, we choose a set of queries $Q_t \in \mathbb{R}^d$ at each round $t \in T$. This leads to an empirical counterpart of our problem, i.e.

$$\max_{\omega \in \omega} \max_{Q_{[0,T]} \in \mathbf{D}^Q_{[T]}} \hat{F}(\omega, Q_{[0,T]}) \triangleq \max_{\omega \in \omega} \max_{Q_{[0,T]} \in \mathbf{D}^Q_{[T]}} -\frac{1}{T} \sum_{t=1}^T l(f_{\theta^*}^T(Q_t), f_\omega^E(Q_t)) + \sum_{t=1}^T H(f_\omega^E(Q_t)).$$

$$(6)$$

As we need to evaluate $f_{\theta^*}^T$ for each $Q_t$, we refer $Q_t$'s as queries, the dataset $\mathbf{D}^Q \subseteq \mathbb{R}^d \times \mathcal{Y}$ from where they are chosen as the query dataset, and the corresponding unobserved distribution $\mathcal{D}^Q$ as the query generating distribution. Given the optimisation problem of Equation 6, we propose an algorithm MARICH to solve it effectively.

## 5 MARICH: A QUERY SELECTION ALGORITHM FOR MODEL EXTRACTION

In this section, we propose an algorithm, MARICH, to solve Equation 6 in a query-efficient manner.

**Algorithm Design.** In Equation 6, we observe that once the queries $Q_{[0,T]}$ are selected the outer maximisation problem, equivalent to regualrised loss minimisation, can be solved using any standard empirical risk minimisation algorithm, such as Adam, SGD etc. Thus, to achieve query efficiency, we focus on designing a query selection algorithm that selects a batch $Q_t$ at round $t \leq T$ such that

$$Q_t \triangleq \arg\max_{Q \in \mathbf{D}^Q} -\frac{1}{t} \sum_{i=1}^{t-1} l(f_{\theta^*}^T(Q_i \cup Q), f_{\omega_{t-1}}^E(Q_i \cup Q))] + \sum_{i=1}^{t-1} H(f_{\omega_{t-1}}^E(Q_i \cup Q)). \tag{7}$$

Here, $f_{\omega_{t-1}}^E$ is the model extracted by round $t-1$. The objective of query selection using Equation 7 is two-fold. First, we want to select a query that maximises the entropy of predictions for the extracted model $f_{\omega_{t-1}}^E$. This allows us to select the queries which are most informative about the mapping between the input features and the prediction space. Secondly, Equation 7 also pushes the adversary to select queries where the target and extracted models mismatch the most. Thus, minimising the loss between target and extracted models for such a query forces them to match over the whole domain of queries. In MARICH, we execute this query selection in two phases.

*Initialisation Phase.* In the Phase 1, i.e. the initialisation phase, we select a set of $n_0$ queries, called $Q_0^{train}$ uniformly randomly from the query dataset $\mathbf{D}^Q$. We send these queries to the target model and collect corresponding predicted classes $Y_0^{train}$ (Line 3). We use these $n_0$ samples of input-predicted label pairs to construct a primary extracted model $f_0^E$.

*Adaptive Phase.* In Phase 2, i.e. the adaptive phase, we select $\gamma_1 \gamma_2 B$ number of queries at every round $t$. In order to be query-efficient, we sequentially deploy three query selection mechanisms, namely ENTROPYSAMPLING, ENTROPYGRADIENTSAMPLING, and LOSSSAMPLING. These techniques together aim to select a batch of $Q_t$ at every step that optimises the objective of Equation 7.

ENTROPYSAMPLING. First, we aim to select the set of queries which unveil most information about the mapping between the input features and the prediction space. Thus, we deploy ENTROPYSAMPLING. In ENTROPYSAMPLING, we compute the output probability vectors from $f_{t-1}^E$ for all the query points in $\mathbf{D}^Q \setminus Q_{t-1}^{train}$ and then select top $B$ points with highest entropy (Line 18). This selects the queries $Q_t^{entropy}$, about which $f_{t-1}^E$ is most confused and training on these essentially would make the model a better classifier.

ENTROPYGRADIENTSAMPLING. To be frugal about the number of queries, we refine $Q_t^{entropy}$ to compute the most diverse subset of it. First, we compute the gradients of entropy of $f_{t-1}^E(x)$, i.e. $\nabla_x H(f_{t-1}^E(x))$, for all $x \in Q_t^{entropy}$. The gradient at point $x$ reflects the change at $x$ in the prediction distribution induced by $f_{t-1}^E$. We use these gradients to embed the points $x \in Q_t^{entropy}$. Now, we deploy K-means clustering to find $k$ (= #classes) clusters and sample $\gamma_1 B$ points (Line 25), namely $Q_t^{grad}$, from these clusters. Selecting points from $k$ different clusters ensures diversity of queries and reduces the number of queries by $\gamma_1$.

LOSSSAMPLING. In this step, we select points from $Q_t^{grad}$ for which the predictions of $f_{\theta^*}^T$ and $f_{t-1}^E$ are most dissimilar. To identify these points, we compute the loss $l(f^T(x), f_{t-1}^E(x))$ for all $x \in Q_{t-1}^{train}$. Then, we select top-$k$ points from $Q_{t-1}^{train}$ with the highest loss values (Line 29), and sample a subset $Q_t^{loss}$ of size $\gamma_1 \gamma_2 B$ from $Q_t^{grad}$ which are closest to the $k$ points selected from $Q_{t-1}^{train}$ (Line 31). This ensures that $f_{t-1}^E$ would align with $f^T$ better if it trains on the points at which the mismatch in their predictions is the highest.

**Remark 2.** *Eq. (7) dictates that the active sampling strategy should try to select queries that maximise the entropy in the prediction distribution of the extracted model, while decreases the mismatch in predictions of the target and the extracted models. We further use the* ENTROPYGRADIENTSAMPLING *to choose a smaller but most diverse subset. As Eq. (7) does not specify any ordering between these objectives, one can argue about the sequence of using these three sampling strategies. We choose to use sampling strategies in the decreasing order of runtime complexity as the first strategy selects the queries from the whole query dataset, while the following strategies work only on the already selected queries. We show in Appendix C that* LOSSSAMPLING *incurs the highest runtime followed by* ENTROPYGRADIENTSAMPLING, *while* ENTROPYSAMPLING *is significantly cheaper.*

---

**Algorithm 1** MARICH

---

**Input**: Target model: $f^T$, Query dataset: $\mathbf{D}^Q$, #Classes: $k$
**Parameter**: #initial samples: $n_0$, Training epochs: $E_{max}$, #Batches of queries: $T$, Query budget: $B$, Subsampling ratios: $\gamma_1, \gamma_2 \in (0, 1]$
**Output**: Extracted model $f^E$

  1: //* Initialisation of the extracted model*//                                                    ▷ Phase 1
  2: $Q_0^{train} \leftarrow n_0$ datapoints randomly chosen from $D^Q$
  3: $Y_0^{train} \leftarrow f^T(Q_0^{train})$                                        ▷ Query the target model $f^T$ with $Q_0^{train}$
  4: **for** epoch $\leftarrow 1$ to $E_{max}$ **do**
  5:     $f_0^E \leftarrow$ Train $f^E$ with $(Q_0^{train}, Y_0^{train})$
  6: //* Adaptive query selection to build the extracted model*//                                 ▷ Phase 2
  7: **for** $t \leftarrow 1$ to $T$ **do**
  8:     $Q_t^{entropy} \leftarrow \text{ENTROPYSAMPLING}(f_{t-1}^E, \mathbf{D}^Q \setminus Q_{t-1}^{train}, B)$
  9:     $Q_t^{grad} \leftarrow \text{ENTROPYGRADIENTSAMPLING}(f_{t-1}^E, Q_t^{entropy}, \gamma_1 B)$
 10:     $Q_t^{loss} \leftarrow \text{LOSSSAMPLING}(f_{t-1}^E, Q_t^{grad}, Q_{t-1}^{train}, Y_{t-1}^{train}, \gamma_1\gamma_2 B)$
 11:     $Y_t^{new} \leftarrow f^T(Q_t^{loss})$                                    ▷ Query the target model $f^T$ with $Q_t^{loss}$
 12:     $Q_t^{train} \leftarrow Q_{t-1}^{train} \cup Q_t^{loss}$
 13:     $Y_t^{train} \leftarrow Y_{t-1}^{train} \cup Y_t^{new}$
 14:     **for** epoch $\leftarrow 1$ to $E_{max}$ **do**
 15:         $f_t^E \leftarrow$ Train $f_{t-1}^E$ with $(Q_t^{train}, Y_t^{train})$
 16: **return** Extracted model $f^E \leftarrow f_T^E$

 17: **function** ENTROPYSAMPLING(extracted model: $f^E$, input data points: $X_{in}$, budget: $B$)
 18:     $Q_{entropy} \leftarrow \arg\max_{X \subset X_{in}, |X|=B} H(f^E(X_{in}))$ ▷ Select $B$ points with maximum entropy
 19:     **return** $Q_{entropy}$

 20: **function** ENTROPYGRADIENTSAMPLING(extracted model: $f^E$, input: $X_{in}$, budget: $\gamma_1 B$)
 21:     $E \leftarrow H(f^E(X_{in}))$
 22:     $G \leftarrow \{\nabla_x E \mid x \in X_{in}\}$
 23:     $C_{in} \leftarrow k$ centres of $G$ computed using K-means
 24:     $Q_{grad} \leftarrow \arg\min_{X \subset X_{in}, |X|=\gamma_1 B} \sum_{x_i \in X} \sum_{x_j \in C_{in}} \|\nabla_{x_i} E - \nabla_{x_j} E\|_2^2$
 25:                                     ▷ Select $\gamma_1 B$ points from $X_{in}$ whose $\frac{\partial E}{\partial x}$ are closest to that of $C_{in}$
 26:     **return** $Q_{grad}$

 27: **function** LOSSSAMPLING(extracted model: $f^E$, input data points: $X_{in}$, previous queries: $Q_{train}$, previous predictions: $Y_{train}$, budget: $\gamma_1\gamma_2 B$)
 28:     $L \leftarrow l(Y_{train}, f^E(Q_{train}))$                                        ▷ Compute the mismatch vector
 29:     $Q_{mis} \leftarrow \text{ARGMAXSORT}(L, k)$                               ▷ Select top-$k$ mismatching points
 30:     $Q_{loss} \leftarrow \arg\min_{X \subset X_{in}, |X|=\gamma_1\gamma_2 B} \sum_{x_i \in X} \sum_{x_j \in Q_{mis}} \|x_i - x_j\|_2^2$
 31:                                                         ▷ Select $\gamma_1\gamma_2 B$ points closest to $Q_{mis}$
 32:     **return** $Q_{loss}$

---

At the end of Phase 2 in each round of sampling, $Q_t^{loss}$ is sent to $f^T$ for fetching the labels $Y_t^{train}$ predicted by the target model. We use $(Q_t^{loss}, Y_t^{loss})$ along with $(Q_{t-1}^{train}, Y_{t-1}^{train})$ to train $f_{t-1}^E$ further. Thus, MARICH performs $n_0 + \gamma_1\gamma_2 BT$ number of queries through $T + 1$ number of interactions with the target model $f^T$ to create the final extracted model $f_T^E$. We describe a pseudocode for MARICH in Algorithm 1. We experimentally demonstrate effectiveness of the model extracted by MARICH to achieve high task accuracy and to act as an informative replica of the target model for extracting private information regarding the private training data $\mathbf{D}^T$.

## 6 EXPERIMENTAL ANALYSIS

In this section, we perform an experimental evaluation of models extracted by MARICH. We discuss the experimental setup, the objectives of experiments, and experimental results. We defer the source code, additional results, effects of defenses, and hyperparameter choices to the appendix.

**Experimental Setup.** We have implemented a prototype of MARICH using Python 3.9 and PyTorch 1.12, and run on a NVIDIA Quadro GV100 32 GB GPU. We perform our attacks against three

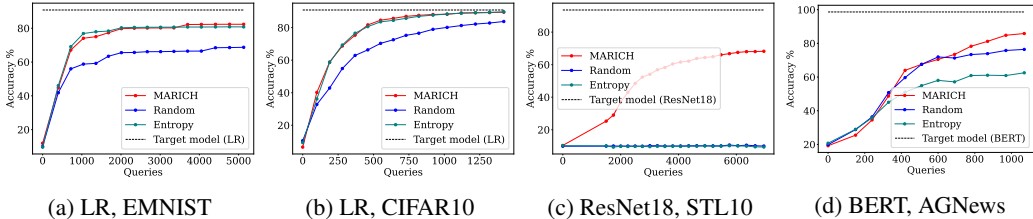

(a) LR, EMNIST      (b) LR, CIFAR10      (c) ResNet18, STL10      (d) BERT, AGNews

Figure 2: Accuracy of the extracted model (mean over 10 runs) in comparison to the target model using MARICH, entropy sampling, and random sampling. Each figure stands for a target model class and a query dataset. Models extracted using queries selected by MARICH are closer to target model.

target models ($f^T$), namely Logistic Regression (LR), ResNet18 (He et al., 2016), and BERT (Devlin et al., 2018), trained on three private datasets ($\mathbf{D}^T$): MNIST handwritten digits (Deng, 2012), CIFAR10 (Krizhevsky et al.) and BBC News, respectively. For model extraction, we use EMNIST letters dataset (Cohen et al., 2017), CIFAR10, STL10, and AGNews (Zhang et al., 2015), as publicly available and mismatched query datasets $\mathbf{D}^Q$. To instantiate task accuracy, we compare accuracy of the extracted models $f^E_{\text{MARICH}}$ with the target model and models extracted by Random Sampling (RS), $f^E_{RS}$. To instantiate informativeness of the extracted models (Nasr et al., 2019), we compare the membership inference (MI), i.e. MI accuracy, AUC of MI, and MI agreements, performed on the target models, and the models extracted using MARICH and RS with same query budget. For MI, we use in-built membership attack from IBM ART (Nicolae et al., 2018). The objectives of our experimental studies are:

1. How do the accuracy of the model extracted using MARICH on the private dataset compare with that of the target model, and RS with same query budget?

2. How do the models extracted by MARICH behave under membership inference attack in comparison to the target model, and the models extracted by RS with same query budget?

**Accuracy of Extracted Models.** MARICH extract logistic regression models with 5,130 and 1,420 queries from EMNIST and CIFAR10 query datasets by attacking a target logistic regression model, $f^T_{logistic}$ trained on MNIST. While the target model achieves 90.82% test accuracy, the models extracted using EMNIST and CIFAR10 achieve test accuracies 82.37% (90.69% of $f^T_{logistic}$) and 89.48% (98.52% of $f^T_{logistic}$), respectively (Figure 2a and 2b). The models extracted using RS show test accuracy 52.96% and 84.18%, and the models extracted by Entropy sampling achieve 80.81% (88.97% of $f^T_{logistic}$) and 89.66% (98.72% of $f^T_{logistic}$). MARICH attacks a ResNet18, $f^T_{ResNet}$ trained on CIFAR10 (accuracy: 93.58%) with 6,950 queries from STL10 dataset. The extracted ResNet18 shows 68.22% (72.90% of $f^T_{ResNet}$) test accuracy. But the model extracted using RS and Entropy sampling achieve 9.99% and 9.39% accuracy (Fig. 2c). To verify MARICH's effectiveness for text data, we attack a BERT, $f^T_{BERT}$ trained on BBC-News (test accuracy: 98.65%) with queries from the AGNews dataset. By using only 1,070 queries, MARICH extracts a model with 87.01% (88.20% of $f^T_{BERT}$) test accuracy (Figure 2d). The model extracted using RS and Entropy sampling shows test accuracy of 76.41% and 62.51%.

Thus, for all the models and datasets, MARICH is able to extract models that achieve closer test accuracy with respect to the target models and are more accurate than the models extracted by both Entropy sampling and RS.

**Membership Inference with Extracted Models.** In Table 1, we report the statistics, i.e. *accuracy*, *agreement* in inference with target model, and *agreement AUC*, of membership attacks performed on different target models and extracted models with different query datasets. The models extracted using MARICH demonstrate higher MI agreement with the target models than the models extracted using RS (Figure 4). They also achieve comparable MI accuracy with respect to the target model. These

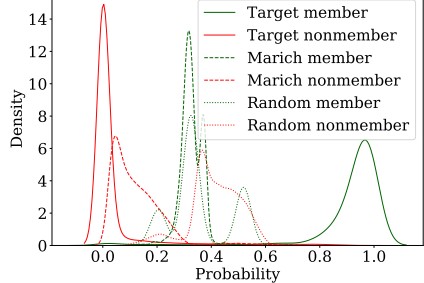

Figure 3: BERT with AGNews queries

results instantiate that the models extracted by MARICH act as informative replicas of the target.

Table 1: Statistics of membership inference (MI) for different target models, datasets & attacks

| Member dataset | Target model | Query Dataset | Algorithm | Non-member dataset | #Queries (% Query Dataset) | MI acc. | MI agreement | MI agreement AUC |
|---|---|---|---|---|---|---|---|---|
| MNIST | LR | - | - | EMNIST | 50,000 (100%) | 87.99% | - | - |
| MNIST | LR | - | - | CIFAR10 | 50,000 (100%) | 92.30% | - | - |
| MNIST | LR | EMNIST | MARICH | EMNIST | 5,130 (3.5%) | 88.58% | 92.82% | **92.73%** |
| MNIST | LR | CIFAR10 | MARICH | CIFAR10 | 1,420 (2.37%) | **94.27%** | **93.97%** | 92.43% |
| MNIST | LR | EMNIST | RS | EMNIST | 5,130 (3.5%) | 89.61% | 91.01% | 91.11% |
| MNIST | LR | CIFAR10 | RS | CIFAR10 | 1,420 (2.37%) | 92.61% | 89.84% | 85.79% |
| CIFAR10 | Resnet18 | - | - | STL10 | 40,000 (100%) | 79.35% | - | - |
| CIFAR10 | Resnet19 | STL10 | MARICH | STL10 | 6,950 (6.15%) | **93.90%** | **75.52%** | **76.69%** |
| CIFAR10 | Resnet19 | STL10 | RS | STL10 | 6,950 (6.15%) | 92.32% | 75.25% | 75.83% |
| BBCNews | BERT | - | - | AGNews | 1,490 (100%) | **98.61%** | - | - |
| BBCNews | BERT | AGNews | MARICH | AGNews | 1,070 (0.83%) | 94.42% | **91.02%** | **82.62%** |
| BBCNews | BERT | AGNews | RS | AGNews | 1,070 (0.83%) | 89.17% | 86.93% | 58.64% |

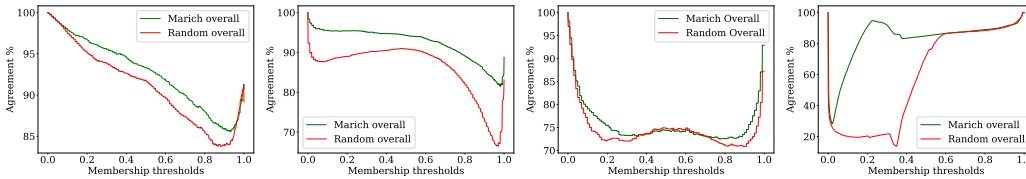

(a) LR with EMNIST    (b) LR with CIFAR10    (c) ResNet18 with STL10    (d) BERT with AGNews

Figure 4: Membership agreements of models extracted by MARICH and by Random Sampling (mean over 10 runs). Each figure stands for a target model class and a query dataset. Models extracted using queries selected by MARICH are higher.

**Analysis of Membership Distributions.** To analyse the outcomes of MI attacks further, we visualise the membership distributions for a BERT model trained on BBCNews dataset, and two BERT models, $f^E_{\text{MARICH}}$ and $f^E_{random}$ extracted using MARICH and RS respectively. To check efficacy of MI, we use BBCNews as the member dataset and AGNews as the non-member dataset. After training 5 neural nets on them, we compute the distribution of membership probabilities for the member and non-member data points for $f^T$, $f^E_{\text{MARICH}}$ and $f^E_{random}$. From Figure 3, we observe that MI on $f^E_{\text{MARICH}}$ produces membership distributions closer to those of $f^T$ than those of $f^E_{random}$.

**Summary of Results.** From the experimental results, we deduce the following conclusions.
*Accuracy.* Test accuracy (on the subsets of private datasets) of the models extracted using MARICH are higher than the models extracted with random queries and ... of the target model (Figure 2 and Table 1). This shows effectiveness of MARICH as a task accuracy extraction attack, while solving distributional equivalence and max-info extractions.
*Effective Membership Inference.* The agreement in MI achieved by attacking $f^E_{\text{MARICH}}$ and the target model is always higher than that of the $f^E_{RS}$ (Figure 4). Also, membership accuracy for $f^E_{\text{MARICH}}$s are $88.58\% - 94.42\%$ (Table 1). This shows that the models extracted by MARICH act as informative replicas of the target model.
*Query-efficiency.* Table 1 shows that MARICH uses only 1,070 - 6,950 queries from the public query datasets, which in most cases are lower than data used for training the target models. This shows MARICH is significantly query efficient whereas other known active learning attacks use at least 10,000 of the query dataset to begin (Pal et al., 2020, Table 2).

# 7 CONCLUSION AND FUTURE DIRECTIONS

In this paper, we investigate the design of a model extraction attack against a target ML model (specifically classifier) trained on a private dataset and accessible through a public API, which returns only a predicted label for a given query. We propose the notions of distributional equivalence extraction, which extends the existing notions task accuracy and functionally equivalent model extractions. We also propose another information-theoretic notion, i.e. Max-Info model extraction. We further propose a variational relaxation of these two types of extraction attacks, and solve it using an online and adaptive query selection algorithm, MARICH. MARICH uses a publicly available query dataset different than the private dataset. We experimentally demonstrate that the models extracted by MARICH achieve $68.22 - 89.48\%$ accuracy on the private dataset while using 1,070 - 6,950 queries. For both text and image data, we demonstrate that the models extracted by MARICH act as informative replicas of the target models leading to $79.35 - 94.42\%$ accuracy and $75.52 - 93.97\%$ agreement in MI on the target model. Typically, the functional equivalence attacks require model-specific techniques, while MARICH is model-oblivious while performing distributional equivalence attack. This poses an interesting question: is distributional equivalence extraction 'easier' than functional equivalence extraction, which is NP-hard (Jagielski et al., 2020)?

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

## A    PROOFS OF SECTION 4

In this section, we elaborate the proofs for the Theorems 1 and 2.[2]

**Theorem 1** (Upper Bounding Distributional Closeness). *If we choose KL-divergence as the divergence function $D$, we can show that*

$$D_{\mathrm{KL}}(\Pr(f_{\theta^*}^T(Q), Q) \| \Pr(f_{\omega_{\mathrm{DEq}}^*}^E(Q), Q)) \leq \min_{\omega} \mathbb{E}_Q[l(f_{\theta^*}^T(Q), f_{\omega}^E(Q))] - H(f_{\omega}^E(Q)).$$

*Proof.* Let us consider a query generating distribution $\mathcal{D}^Q$ on $\mathbb{R}^d$. A target model $f_{\theta^*}^T : \mathbb{R}^d \to \mathcal{Y}$ induces a joint distribution over the query and the output (or label) space, denoted by $\Pr(f_{\theta^*}^T, Q)$. Similarly, the extracted model $f_{\theta^*}^T : \mathbb{R}^d \to \mathcal{Y}$ also induces a joint distribution over the query and the output (or label) space, denoted by $\Pr(f_{\omega}^E, Q)$.

$$D_{\mathrm{KL}}(\Pr(f_{\theta^*}^T(Q), Q) \| \Pr(f_{\omega}^E(Q), Q))$$

$$= \int_{Q \in \mathbb{R}^d} \mathrm{d} \Pr(f_{\theta^*}^T(Q), Q) \log \frac{\Pr(f_{\theta^*}^T(Q), Q)}{\Pr(f_{\omega}^E(Q), Q)}$$

$$= \int_{Q \in \mathbb{R}^d} \Pr(f_{\theta^*}^T(Q)|Q = q) \Pr(Q = q) \log \frac{\Pr(f_{\theta^*}^T(Q)|Q = q)}{\Pr(f_{\omega}^E(Q)|Q = q)} \, \mathrm{d}q$$

$$= \int_{Q \in \mathbb{R}^d} \Pr(f_{\theta^*}^T(Q)|Q = q) \Pr(Q = q) \log \Pr(f_{\theta^*}^T(Q)|Q = q) \, \mathrm{d}q$$

$$- \int_{Q \in \mathbb{R}^d} \Pr(f_{\theta^*}^T(Q)|Q = q) \Pr(Q = q) \log \Pr(f_{\omega}^E(Q)|Q = q) \, \mathrm{d}q$$

$$= \int_{Q \in \mathbb{R}^d} \Pr(f_{\theta^*}^T(Q)|Q = q) \Pr(Q = q) \log \Pr(f_{\theta^*}^T(Q)|Q = q) \, \mathrm{d}q + \mathbb{E}_{q \sim \mathcal{D}^Q} \left[ l(f_{\theta^*}^T(q)), f_{\omega}^E(q)) \right]$$

$$\leq -H(f_{\theta^*}^T(Q) \, \mathrm{d}q + \mathbb{E}_{q \sim \mathcal{D}^Q} \left[ l(f_{\theta^*}^T(q)), f_{\omega}^E(q)) \right]$$

$$\leq -H(f_{\omega}^E(Q) \, \mathrm{d}q + \mathbb{E}_{q \sim \mathcal{D}^Q} \left[ l(f_{\theta^*}^T(q)), f_{\omega}^E(q)) \right] \tag{8}$$

The last inequality holds true as the extracted model $f_{\omega}^E$ is trained using the outputs of the target model $f_{\theta^*}^T$. Thus, by data-processing inequality, its output distribution posses less information than that of the target model. Specifically, we know that if $Y = f(X)$, $H(Y) \leq H(X)$.

Now, by taking $\min_{\omega}$ on both sides, we obtain

$$D_{\mathrm{KL}}(\Pr(f_{\theta^*}^T(Q), Q) \| \Pr(f_{\omega_{\mathrm{DEq}}^*}^E(Q), Q)) \leq \min_{\omega} \mathbb{E}_Q[l(f_{\theta^*}^T(Q), f_{\omega}^E(Q))] - H(f_{\omega}^E(Q)).$$

Here, $\omega_{\mathrm{DEq}}^* \triangleq \arg\min_{\omega} D_{\mathrm{KL}}(\Pr(f_{\theta^*}^T(Q), Q) \| \Pr(f_{\omega}^E(Q), Q))$. The equality exists if minima of LHS and RHS coincide. □

**Theorem 2** (Lower Bounding Information Leakage). *The information leaked by any max-information attack (Equation 3) is lower bounded as follows:*

$$I(\Pr(f_{\theta^*}^T(Q), Q) \| \Pr(f_{\omega_{\mathrm{MaxInf}}^*}^E(Q), Q)) \geq \max_{\omega} -\mathbb{E}_Q[l(f_{\theta^*}^T(Q), f_{\omega}^E(Q))] + H(f_{\omega}^E(Q)).$$

*Proof.* Let us consider the same terminology as the previous proof. Then,

$$I(\Pr(f_{\theta^*}^T(Q), Q) \| \Pr(f_{\omega}^E(Q), Q))$$

$$= H(f_{\theta^*}^T(Q), Q) + H(f_{\omega}^E(Q), Q) - H(f_{\theta^*}^T(Q), f_{\omega}^E(Q), Q)$$

$$= H(f_{\theta^*}^T(Q), Q) + H(f_{\omega}^E(Q), Q) - H(f_{\omega}^E(Q), Q|f_{\theta^*}^T(Q)) + H(f_{\theta^*}^T(Q))$$

$$\geq H(f_{\omega}^E(Q), Q) - H(f_{\omega}^E(Q), Q|f_{\theta^*}^T(Q)) \tag{9}$$

$$\geq H(f_{\omega}^E(Q)) - H(f_{\omega}^E(Q), Q|f_{\theta^*}^T(Q)) \tag{10}$$

$$\geq H(f_{\omega}^E(Q)) - \mathbb{E}_Q[l(f_{\omega}^E(Q), f_{\theta^*}^T(Q))] \tag{11}$$

---

[2]Throughout the proofs, we slightly abuse the notation to write $l(\Pr(X), \Pr(Y))$ as $l(X, Y)$ for avoiding cumbersome equations.

The inequality of Equation 9 is due to the fact that entropy is always non-negative. Equation 10 hols true as $H(X, Y) \geq \max\{H(X), H(Y)\}$ for two random variables $X$ and $Y$. The last inequality is due to the fact that conditional entropy of two random variables $X$ and $Y$, i.e. $H(X|Y)$, is smaller than or equal to their cross entropy, i.e. $l(X, Y)$ (Lemma 1).

Now, by taking $\max_\omega$ on both sides, we obtain

$$I(\Pr(f_{\theta^*}^T(Q), Q) \| \Pr(f_{\omega_{\text{MaxInf}}^*}^E(Q), Q)) \leq \max_\omega -\mathbb{E}_Q[l(f_{\theta^*}^T(Q), f_\omega^E(Q))] + H(f_\omega^E(Q)).$$

Here, $\omega_{\text{MaxInf}}^* \triangleq \arg\max_\omega I(\Pr(f_{\theta^*}^T(Q), Q) \| \Pr(f_{\omega_{\text{MaxInf}}^*}^E(Q), Q))$. The equality exists if maxima of LHS and RHS coincide. $\qquad\square$

**Lemma 1** (Relating Cross Entropy and Conditional Entropy). *Given two random variables $X$ and $Y$, conditional entropy*

$$H(X|Y) \leq l(X, Y). \tag{12}$$

*Proof.* Here, $H(X|Y) \triangleq -\int \Pr(x, y) \log \frac{\Pr(x,y)}{\Pr(y)}$ and $l(X, Y) \triangleq l(\Pr(X), \Pr(Y)) = -\int \Pr(x) \ln \Pr(y)$ denotes the cross-entropy.

$$\begin{aligned}
l(X, Y) &= H(X) + D_{\text{KL}}(\Pr(X) \| \Pr(Y)) \\
&= H(X|Y) + I(X; Y) + D_{\text{KL}}(P_X \| P_Y) \\
&\geq H(X|Y)
\end{aligned}$$

The last inequality holds as both mutual information $I$ and KL-divergence $D_{\text{KL}}$ are non-negative functions for any $X$ and $Y$. $\qquad\square$

## B  ADDITIONAL EXPERIMENTAL RESULTS

In this section, we elaborate further experimental setups that we skipped for the brevity of space in the main draft. We provide an anonymised version of the code at: `https://drive.google.com/drive/folders/1mpM-zE3w_pIS0c3DDb_uiR9Jw_MYvVer?usp=sharing`.

### B.1  ACCURACY OF MODELS EXTRACTED BY MARICH AND OTHER SAMPLING STRATEGIES

To compare MARICH with other active learning algorithms, we attack the same target models using Entropy sampling, K-centre sampling, and Random sampling using the same number of queries as used for MARICH.

On the LR models we have shown performances of both Entropy sampling and K-centre sampling, while due to time and resource constraint, we could not present the K-centre sampling results for the BERT and ResNet18.

From the results in Figure 5, we see that in most cases MARICH outperforms other algorithms. In Figure 2, we have not plotted the standard deviations for better visibility. We plot both mean ± standard deviation over 10 runs in Figure 5

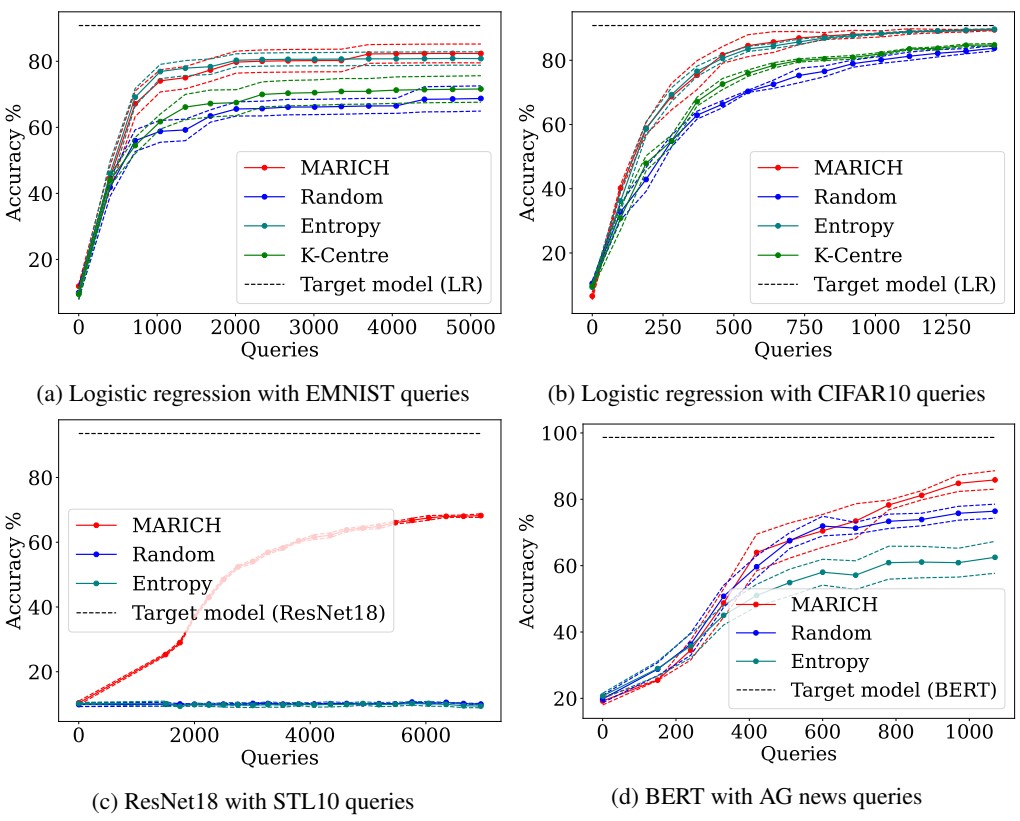

(a) Logistic regression with EMNIST queries     (b) Logistic regression with CIFAR10 queries

(c) ResNet18 with STL10 queries     (d) BERT with AG news queries

Figure 5: Comparison among different active sampling algorithms.

**Extraction of a CNN Trained on MNIST.** Along with the other experiments mentioned in the paper, we trained a CNN with MNIST handwritten digits ($\mathbf{D}^T$ here), that shows a test accuracy of $97.53\%$ on a disjoint test set. Two datasets are used as $\mathbf{D}^Q$ here, EMNIST letters and CIFAR10 to extract two CNN models from the target model. MARICH extracts two CNNs using queries from EMNIST letters and CIFAR10 which show test accuracies of $88.81\%$ and $87.16\%$ respectively. On the other hand, models extracted using random sampling using EMNIST letters and CIFAR10 show accuracies of $85.37\%$ and $88.46\%$ respectively. Table 2 contains queries used, and membership inference statistics for all the experiments.

Table 2: Model extraction and membership inference statistics

| Member dataset | Target model | Attack Dataset | Algorithm used | Non-member dataset | Queries | Membership acc | Nonmembership acc | Overall membership acc | Overall Membership agreement | Membership agreement AUC |
|---|---|---|---|---|---|---|---|---|---|---|
| MNIST | LR | - | - | EMNIST | 50,000 (100%) | 94.91% | 67.24% | 87.99% | - | - |
| MNIST | LR | - | - | CIFAR10 | 50,000 (100%) | 97.13% | 77.80% | 92.30% | - | - |
| MNIST | LR | EMNIST | MARICH | EMNIST | 5,130 (3.5%) | 95.16% | 68.84% | 88.58% | 92.82% | 92.72% |
| MNIST | LR | CIFAR10 | MARICH | CIFAR10 | 1,420 (2.37%) | 97.98% | 83.16% | 94.27% | 93.97% | 92.27% |
| MNIST | LR | EMNIST | Random Sampling | EMNIST | 5,130 (3.5%) | 95.57% | 71.72% | 89.61% | 91.01% | 91.04% |
| MNIST | LR | CIFAR10 | Random Sampling | CIFAR10 | 1,420 (2.37%) | 95.15% | 84.98% | 92.61% | 89.84% | 85.54% |
| MNIST | CNN | - | - | EMNIST | 50,000 (100%) | 91.82% | 74.84% | 87.57% | - | - |
| MNIST | CNN | - | - | CIFAR10 | 50,000 (100%) | 94.94% | 83.05% | 91.97% | - | - |
| MNIST | CNN | EMNIST | MARICH | EMNIST | 5,440 (3.73%) | 95.07% | 80.32% | 91.38% | 92.64% | 91.23% |
| MNIST | CNN | CIFAR10 | MARICH | CIFAR10 | 5545 (9.24%) | 93.48% | 78.66% | 89.78% | 94.92% | 92.27% |
| MNIST | CNN | EMNIST | Random Sampling | EMNIST | 5,440 (3.73%) | 95.95% | 79.50% | 91.84% | 92.35% | 91.11% |
| MNIST | CNN | CIFAR10 | Random Sampling | CIFAR10 | 5545 (9.24%) | 97.59% | 85.90% | 94.66% | 94.16% | 91.35% |
| CIFAR10 | Resnet18 | - | - | STL10 | 40,000 (100%) | 81.55% | 77.15% | 79.35% | - | - |
| CIFAR10 | Resnet19 | STL10 | MARICH | STL10 | 6,950 (6.15%) | 92.80% | 91.75% | 92.32% | 75.52% | 76.36% |
| CIFAR10 | Resnet19 | STL10 | Random Sampling | STL10 | 6,950 (6.15%) | 92.75% | 95.05% | 93.90% | 75.25% | 74.86% |
| BBCNews | BERT | - | - | AGNews | 1,490 (100%) | 91.69% | 99.56% | 98.61% | - | - |
| BBCNews | BERT | AGNews | MARICH | AGNews | 1,070 (0.83%) | 80.96% | 96.25% | 94.42% | 91.02% | 82.16% |
| BBCNews | BERT | AGNews | Random Sampling | AGNews | 1,070 (0.83%) | 23.01% | 98.28% | 89.17% | 86.93% | 58.64% |

## B.2 MEMBERSHIP INFERENCE WITH THE EXTRACTED MODELS

From Figure 6, we see that in most cases the probability densities of the membership inference are closer to the target model when the model is extracted using MARICH, than using random sampling (RS).

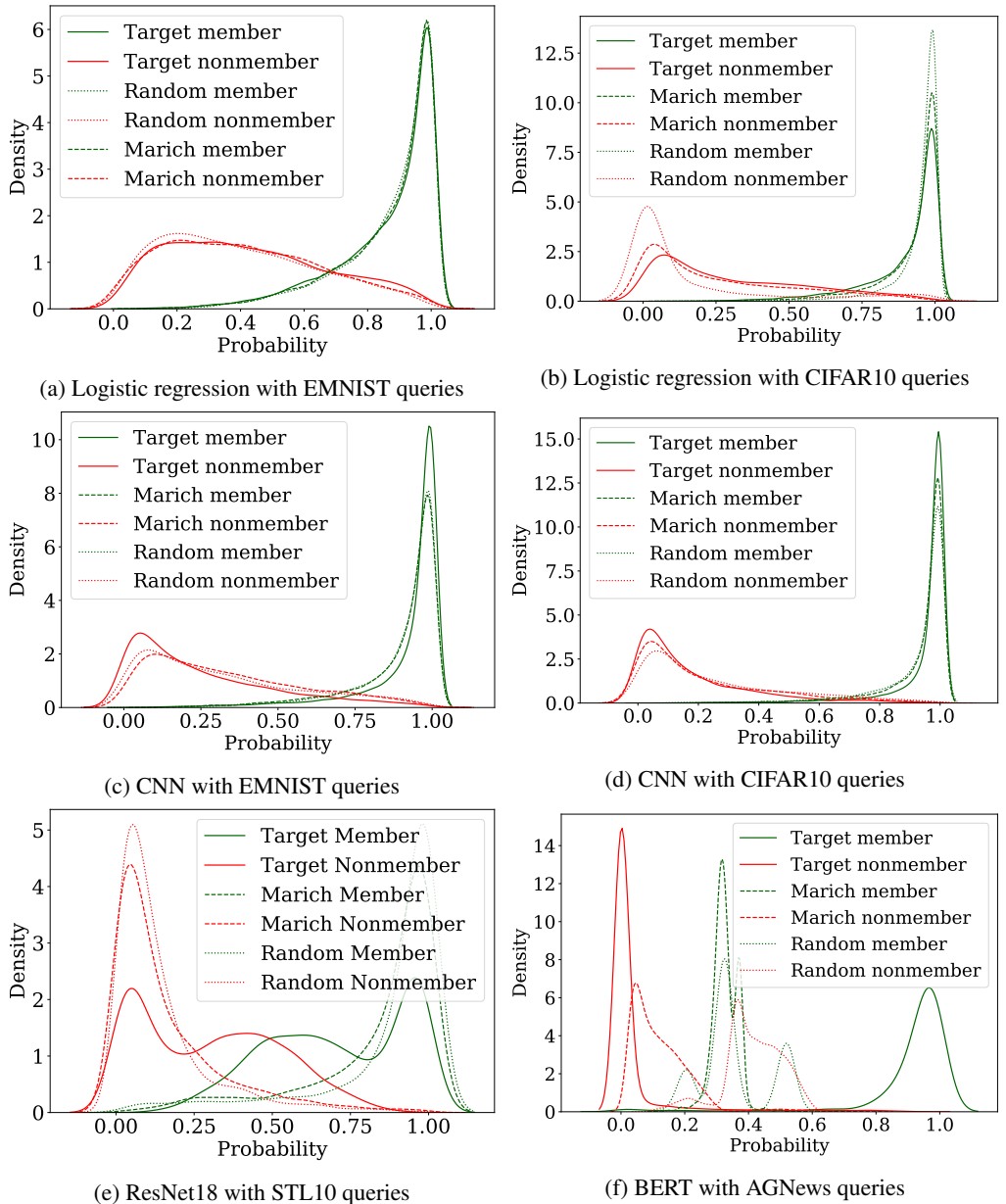

(a) Logistic regression with EMNIST queries

(b) Logistic regression with CIFAR10 queries

(c) CNN with EMNIST queries

(d) CNN with CIFAR10 queries

(e) ResNet18 with STL10 queries

(f) BERT with AGNews queries

Figure 6: Comparison among membership vs. non-membership probability densities for membership attacks against models extracted by MARICH, Random sampling and the target model. Each figure represents the model class and query dataset. Memberships and non-memberships inferred from the model extracted by MARICH are significantly closer to the target model.

In Figure 7, we present the agreements from the member points, nonmember points and overall agreement curves for varying membership thresholds, along with the AUCs of the overall membership agreements. We see that in most cases, the agreement curves for the models extracted using MARICH are above those for the models extracted using random sampling, thus AUCs are higher for the models extracted using MARICH.

These observations support our claim that model extraction using MARICH gives models are accurate and informative replica of the target model.

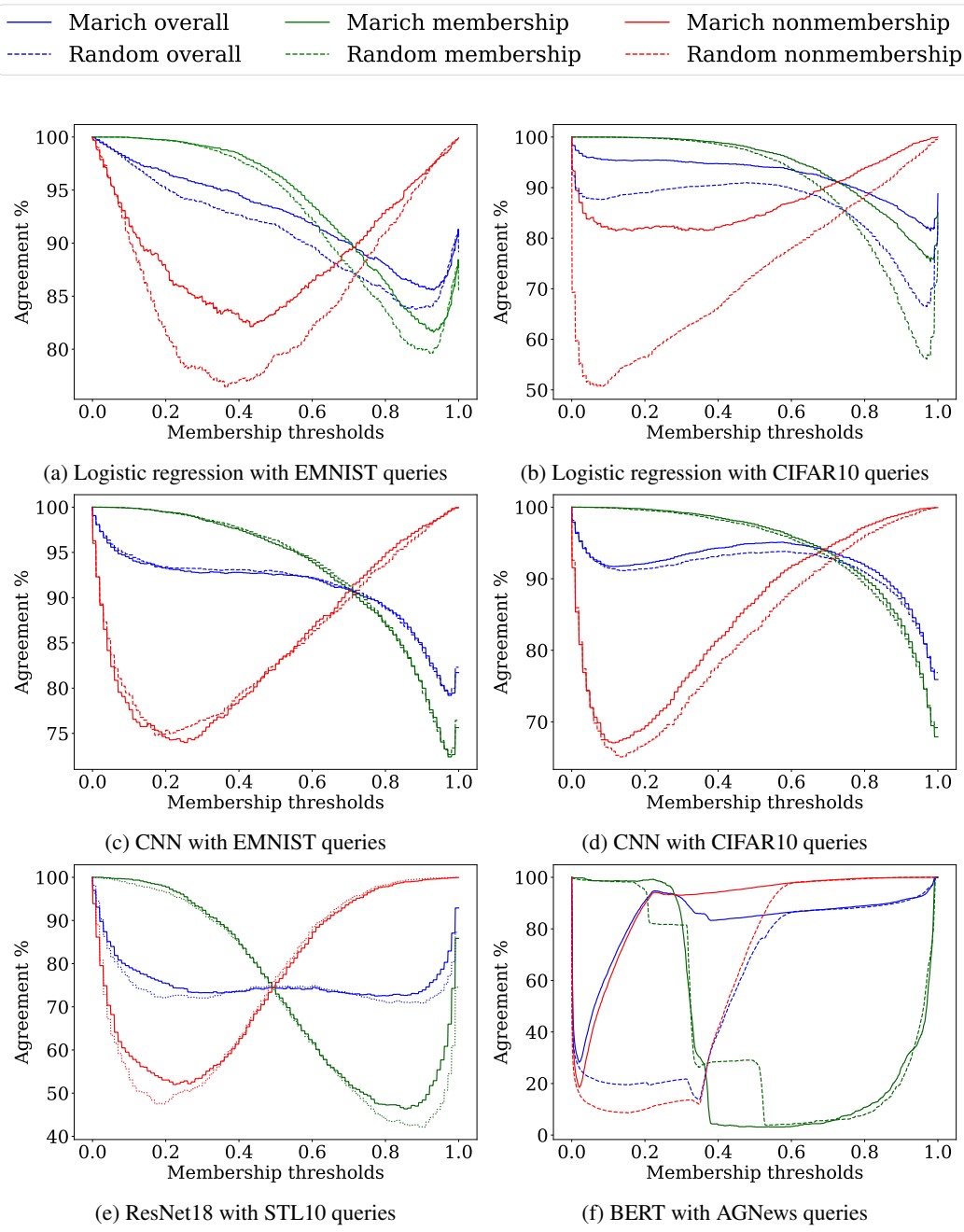

Figure 7: Comparison of membership, nonmembership and overall agreements of membership attacks against models extracted by MARICH and Random sampling and the target model trained with MNIST. Each figure represents the model class and query dataset. Membership agreement of the models extracted by MARICH are higher.

### B.3 FIDELITY OF THE PREDICTION DISTRIBUTIONS OF THE EXTRACTED MODELS

We claim to achieve distributionally equivalent $f^E$ from $f^T$ using MARICH. To measure the performance of MARICH on this objective, we measure KL divergence of the output distributions of $f^T$ and $f^E$ when the input is $\mathbf{D}^Q$ after every round of training.

In Figure 8 we observe that MARICH and Entropy sampling show almost same performance for this particular case while K-centre sampling performs much worse.

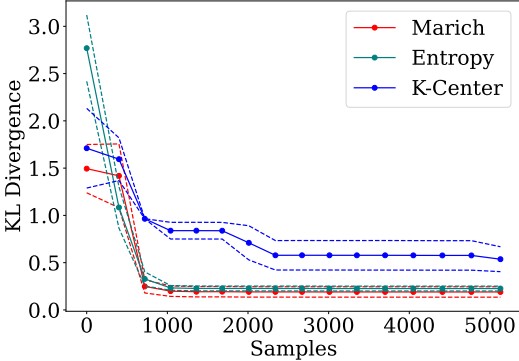

Figure 8: Comparison of fidelity of distributions for different active learning algorithms (LR trained on MNIST attacked with EMNIST).

## C    SIGNIFICANCE AND COMPARISON OF SAMPLING STRATEGIES

Given the bi-level optimization problem, we came up with MARICH in which three sampling methods are used in the order: (i) ENTROPYSAMPLING, (ii) ENTROPYGRADIENTSAMPLING, and (iii) LOSSSAMPLING.

These three sampling techniques contribute to different goals:

- ENTROPYSAMPLING selects points about which the classifier at a particular time step is most confused
- ENTROPYGRADIENTSAMPLING uses gradients of entropy of outputs of the extracted model w.r.t. the inputs as embeddings and selects points behaving most diversely at every time step.
- LOSSSAMPLING selects points which produce highest loss when loss is calculated between target model's output and extracted model's output.

One can argue that the order is immaterial for the optimization problem. But looking at the algorithm practically, we see that ENTROPYGRADIENTSAMPLING and LOSSSAMPLING incur much higher time complexity than ENTROPYSAMPLING. Thus, using ENTROPYSAMPLING on the entire query set is more efficient than the others. This makes us put ENTROPYSAMPLING as the first query selection strategy.

As per the optimization problem in Equation (7), we are supposed to find points that show highest mismatch between the target and the extracted models after choosing the query subset maximising the entropy. This leads us to the idea of LOSSSAMPLING. But as only focusing on loss between models may choose points from one particular region only, and thus, decreasing the diversity of the queries. We use ENTROPYGRADIENTSAMPLING before LOSSSAMPLING. This ensures selection of diverse points with high performance mismatch.

In Figure 9, we experimentally see the time complexities of the three components used. These are calculated by applying the sampling algorithms on a logistic regression model, on mentioned slices of MNIST dataset.

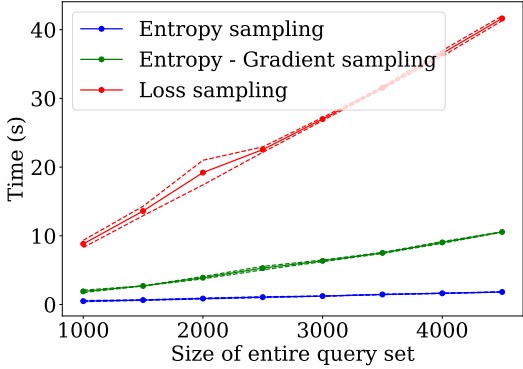

Figure 9: Runtime comparison of three sampling strategies to select queries from 4500 datapoints.

Table 3: Time complexity of different sampling Strategies

| Sampling Algorithm | Query space size | #Selected queries | Time (s) |
|---|---|---|---|
| Entropy Sampling | 4500 | 100 | $1.82 \pm 0.04$ |
| Entropy-Gradient Sampling | 4500 | 100 | $10.56 \pm 0.07$ |
| Loss Sampling | 4500 | 100 | $41.64 \pm 0.69$ |

# D PERFORMANCE AGAINST DIFFERENTIALLY PRIVATE TARGET MODELS

In this section, we aim to verify performance of MARICH against privacy-preserving mechanisms.

Specifically, we apply a $(\varepsilon, \delta)$-Differential Privacy (DP) inducing mechanism, namely DP-SGD (**?**), to train the target model on the member dataset. This mechanism adds noise to the gradients and clip them while training the target model. We use the default implementation of Opacus (Yousefpour et al., 2021) to conduct the training in PyTorch.

Following that, we attack the $(\varepsilon, \delta)$-DP target models using MARICH and compute the corresponding accuracy of the extracted models. In Figure 10, we show the effect of different privacy levels $\varepsilon$ on the achieved accuracy of the extracted Logistic Regression model trained with MNIST dataset and queried with EMNIST dataset. Specifically, we assign $\delta = 10^{-5}$ and vary $\varepsilon$ in $\{0.2, 0.5, 1, 2, \infty\}$. Here, $\varepsilon = \infty$ corresponds to the model extracted from the non-private target model.

We observe that the accuracy of the models extracted from private target models are approximately $2.3 - 7.4\%$ lower than the model extracted from the non-private target model. This shows that performance of MARICH decreases against DP defenses but not significantly.

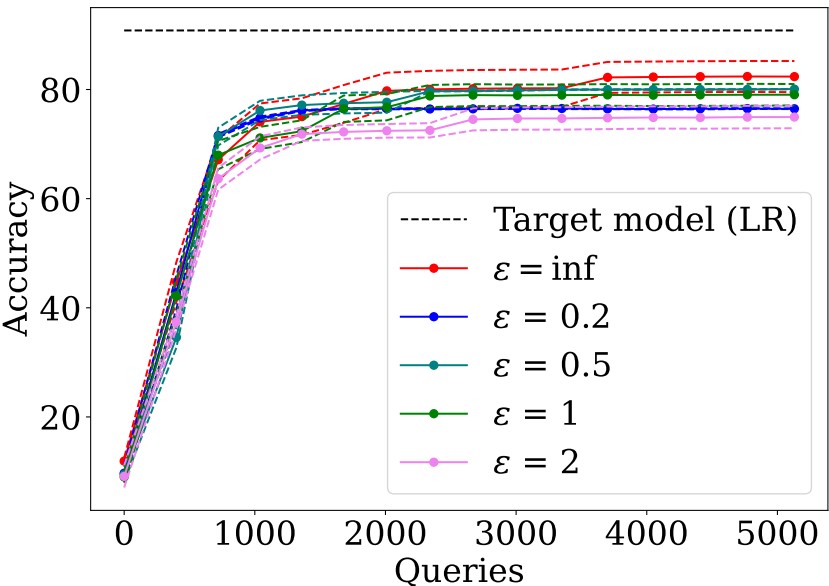

Figure 10: Performance of models extracted by MARICH against $(\varepsilon, \delta)$-differentially private target models for different privacy levels $\varepsilon$ and $\delta = 10^{-5}$.

# E   EFFECT OF MODEL MISMATCH

From Equation equation 7, we observe that functionality of MARICH is not constrained by selecting the same model class for both the target model $f^T$ and the extracted model $f^E$. But in all the previous experiments, we have used the same model class for both the target and extracted models, i.e., we have used LR to extract LR or CNN to extract CNN. In this section, we conduct experiments to show MARICH's capability to handle model mismatch and impact of model mismatch on performance of the extracted models.

Specifically, we run experiments for two cases. We train an LR and a CNN model on MNIST dataset, and use them as target models. We further extract these two models with two other LR and CNN models using EMNIST as the query datasets. We use MARICH without any modification for both the cases when the model classes match and mismatch. This shows universality of MARICH as a model extraction attack.

From Figure 11, we observe that model mismatch influences performance of the model extracted by MARICH. When we extract the LR target model with LR and CNN, we observe that both the extracted models achieve almost same accuracy and the extracted CNN model achieves even a bit more accuracy than the extracted LR model. In contrast, when we extract the CNN target model with LR and CNN, we observe that the extracted LR models achieves lower accuracy than the extracted CNN model.

From these observations, we conclude that if we use a less complex model to extract a more complex model, the accuracy drops significantly. But if we extract a low complexity model with a higher complexity one, we obtain higher accuracy instead of model mismatch. This is intuitive as the low-complexity extracted model might have lower representation capacity to mimic the non-linear decision boundary of the high-complexity model but the opposite is not true.

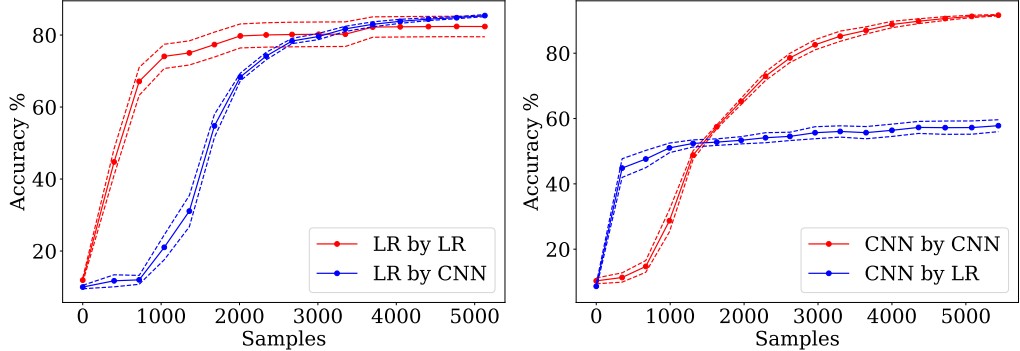

(a) LR extracted by LR vs. LR extracted by CNN   (b) CNN extracted by CNN vs. CNN extracted by LR

Figure 11: Effect of model mismatches on MARICH.

Table 4: Effect of Model mismatch on Accuracy of The Extracted Models.

| $f^E$ | $f^T$ | #samples | Accuracy |
|-------|-------|----------|----------|
| LR | LR | 5130 | $82.37 \pm 5.7\%$ |
| LR | CNN | 5130 | $85.41 \pm 0.57\%$ |
| CNN | LR | 5440 | $57.81 \pm 3.64\%$ |
| CNN | CNN | 5440 | $91.63 \pm 0.42\%$ |

# F CHOICES OF HYPERPARAMETERS

In this section, we list the choices of the hyperparameters of Algorithm 1 for different experiments and also explain how do we select them.

Hyperparameters $\gamma_1$ and $\gamma_2$ are kept constant, i.e., 0.8, for all the experiments. These two parameters act as the budget shrinking factors.

Instead of changing these two, we change the number of points $n_0$, which are randomly selected in the beginning, and the budget $B$ for every step. We obtain the optimal hyperparameters for each experiment by performing a line search in the interval $[100, 500]$.

We further change the budget over the rounds. At time step $t$, the budget, $B_t = \alpha^t \times B_{t-1}$. The idea is to select more points as $f^E$ goes on reaching the performance of $f^T$. Here, $\alpha > 1$ and needs to be tuned. We use $\alpha = 1.02$, which is obtained through a line search in $[1.01, 1.99]$.

For number of rounds $T$, we perform a line search in $[10, 20]$.

Table 5: Hyperparameters for different datasets and target models.

| Member dataset | Target model | Attack dataset | Budget | Initial points | $\gamma_1$ | $\gamma_2$ | Rounds | Epochs/Round | Learning Rate |
|---|---|---|---|---|---|---|---|---|---|
| MNIST | LR | EMNIST | 500 | 400 | 0.8 | 0.8 | 14 | 20 | $2 \times 10^{-2}$ |
| | LR | CIFAR10 | 150 | 100 | 0.8 | 0.8 | 14 | 20 | $2 \times 10^{-2}$ |
| | CNN | EMNIST | 500 | 350 | 0.8 | 0.8 | 15 | 20 | $1 \times 10^{-2}$ |
| CIFAR10 | ResNet18 | STL10 | 390 | 1500 | 0.8 | 0.8 | 20 | 4 | $4 \times 10^{-4}$ |
| BBC News | BERT | AG News | 150 | 150 | 0.8 | 0.8 | 10 | 3 | $2 \times 10^{-6}$ |

