# OpenReview forum: "Marich: A Query-efficient & Online Model Extraction Attack using Public Data"
_ICLR.cc/2023/Conference — Submitted to ICLR 2023_

### Official Review · Reviewer_BNEH · 2022-10-17

**Confidence:** 4
**Correctness:** 2
**Technical Novelty And Significance:** 2
**Empirical Novelty And Significance:** 2
**Recommendation:** 3

**Clarity, Quality, Novelty And Reproducibility:**

Neat:

- Introduction: a user and not an user
- At the end of the introduction, the 3rd point - The percentage of data points is not informative - I suppose you wanted to add the % of the training samples used by the victim.
- Grammar: (1) Point 2 in the introduction: sendS the queries, collectS, and useS. (2) Section 3 - adversary aimS at ... and createS another replica
- Figure 2 - the legend is way too small.

**Strength And Weaknesses:**

Comments:

- The paper considers the model extraction attack from a theoretical perspective and tries to optimize the query selection.
- The background on classifiers is totally unnecessary, at the beginning of Section 3, page 3.
- Merge part of Section 2 on Related Work "Taxonomy of Model Extraction" with parts of Section 3: "Model Extraction Attack" - this is too repetitive.
- The formalism of model extraction in Section 4 "DISTRIBUTIONAL EQUIVALENCE & MAX-INFORMATION MODEL EXTRACTIONS: A VARIATIONAL OPTIMISATION FORMULATION" is not informative - it only formalizes model stealing while it does not provide any insight on the model extraction itself. Most of this Section could be moved to the appendix.
- The ideas behind Marich: (1) max entropy of the predictions for the extracted model, and (2) an adversary selects queries where the target and extracted models mismatch the most - are very intuitive and known without a formal derivation. The 1st one is the principle behind active learning while the 2nd naturally evolves from fidelity-driven extraction.
- Even in the initialization phase of Marich, you could select the most diverse set of queries - these whose, e.g. L2 distance between each other is the largest.
- Note, you still select queries for stealing that are close in terms of their distribution to the training samples of the victim models.


Questions:

- How do the samplings: of entropy, gradient, and loss reflect the optimization stated in Equation 7?
- Why does NOT the random selection of queries (the red line) in Figure 2 start from 0 and goes up with more queries sent against the public API?
- On page 8, what is the meaning behind the percentage: "with 6,950 (6.15%) queries from STL10 dataset"?
- I assume the authors do not train the BERT model from scratch on the BBCNews data but only fine-tuned a pre-trained BERT. Which exactly BERT model was used?
- Why does the model extracted using random selection achieves only 24.44% - "the model extracted using RS achieves 24.44% (26.12% of fTResNet) accuracy (Fig. 2c)." The provided information: "0.83 − 6.15% of the query dataset" is not giving any valuable insight since the query dataset can be arbitrarily large.

Related work
1. Type of Query dataset from page 3 - the data-free model extraction generates queries using a GAN (Truong et al. 2021).
2. The related work is incomplete - check the latest approaches, for instance: https://openreview.net/forum?id=EAy7C1cgE1L
3. This paper cites Jagielski et al. but does not compare with any other attacks, such as MixMatch - proposed by Jagielski et al. or does not compare with baselines that use active learning (e.g., via entropy sampling, k-centers, or margin sampling). The attacks like KnockOff (Orekondy et al. 2020) that also leverages active learning (Chandrasekaran et al. 2020) or defenses that use information leakage via privacy leakage, margin sampling, or entropy (Dziedzic et al. 2022), are not even discussed.



**Summary Of The Paper:**

The paper proposes Marich - a model extraction attack with an adaptive query selection. The three main contributions are formalism, algorithm, and experimental analysis. The authors formally formulate the problem as the Max-Information attack, where the adversary aims to maximize the mutual information between the extracted and target models’ distributions. The proposed algorithm Marich - optimizes the objective of the variational optimization problem. The experimental evaluation is done with both image and text datasets as well as on standard vision (ResNet18) and NLP (BERT) architecture types.

**Summary Of The Review:**

The paper does not provide any new important insights into the problem of model extraction. The comparison with baselines is missing.

---

### Official Review · Reviewer_LzQT · 2022-10-20

**Confidence:** 4
**Correctness:** 2
**Technical Novelty And Significance:** 2
**Empirical Novelty And Significance:** 1
**Recommendation:** 3

**Clarity, Quality, Novelty And Reproducibility:**

The paper is not well-written and the ideas are hard to follow.
Besides, the proposed method seems to be a combination of existing approaches, which lack novelty.

**Strength And Weaknesses:**

In general, I think this paper is written badly and the proposed method seems to be trivial. Detailed comments are listed below:

1. The ideas in Section 4 are very hard to follow. In this Section, the authors seem to propose two new definitions (Def. 1 and Def. 2) of model extraction performance indicators, but I could not figure out the relationship between these definitions and the proposed MARICH algorithm.

2. In Eq.(1), the authors claim that "an adversary should choose a query generating distribution $\mathcal D^Q$ that can minimize the difference between the target model and extracted model". However, I do not agree with that.
On the one hand, if we let $\mathcal D^Q$ only generate zero values, it would probably result in a small distributional model difference but not help extract informative knowledge from the target model.
On the other hand, from my perspective, a query generating distribution that can result in large differences between extracted and target models may help extract more information from the victims, as larger model prediction disagreement may suggest more knowledge to be transferred by query data.

3. The proposed MARICH algorithm seems to be a combination of several existing heuristic strategies that have been already studied in model stealing and other aspects of ML security. Specifically:
    - The entropy-sampling is as same as that of the "Uncertainty Strategy" in ActiveTheif method [r1].
    - For the gradient-sampling, a similar idea of leveraging gradient information has been used in [r2] for detecting backdoored data.
    - The loss-sampling is an (slightly) improved "DFAL + k-center strategy" in ActiveTheif method [r1].

    All these results may shrink the contribution of the proposed MARICH method.

4. Too few model extraction attack baselines for comparison. The authors only compare MARICH with the random sampling-based method. However, I think they should at least include [r1], which is another existing active learning-based model extraction method, for comparison.


[r1] Pal et al. "ActiveThief: Model Extraction Using Active Learning and Unannotated Public Data". AAAI 2020.

[r2] Xu et al. "Detecting AI Trojans Using Meta Neural Analysis". IEEE SP 2021.

**Summary Of The Paper:**

This paper studies how to perform model extraction attacks in a query-efficient manner. Specifically, the authors design a new extraction method named MARICH that leverages active learning to select informative query data to improve the query efficiency of model extraction. Three active learning strategies, entropy-sampling, gradient-sampling, and loss-sampling, are deployed sequentially to select the most informative query data. Experiments are conducted on image and text datasets to verify the effectiveness of the MARICH, in which statistics of classification accuracies and membership inference statistics are employed to evaluate the performance of model extraction attacks.

**Summary Of The Review:**

I think this paper should be rejected for now.

---

### Official Review · Reviewer_4xVs · 2022-10-26

**Confidence:** 3
**Correctness:** 3
**Technical Novelty And Significance:** 3
**Empirical Novelty And Significance:** 3
**Recommendation:** 6

**Clarity, Quality, Novelty And Reproducibility:**

**Clarity**: Very good. I was able to follow most details in the paper. The paper also provides a reasonable background and landscape of related work for new readers.

**Quality**: Average. The technical contributions appear relevant (e.g., distributional equivalence notion, query-efficient strategy). The only issue I find is that the introduction slightly appears to overclaim (the paper in the end target task accuracy) and the evaluation has some gaps to confidently support the query-efficient claims.

**Novelty**: Good. The paper introduces a reasonable distributional equivalence framework and proposes a query-efficient approach with significant improvements over random sampling.

**Reproducibility**: Poor. I could not find many implementation details and I'm not confident in reproducing the results. (See concern #1 for more details)

**Strength And Weaknesses:**

### Strengths
**1. Formal framework**
- I appreciate the paper's contribution on unifying a number of recent approaches within a reasonable framework (Eq. 6). While most papers inherently solve this bi-level optimization problem, this is the first paper I'm aware of presents the extraction formulation and further grounds it in a variational formulation.

**2. Significant query-efficient improvements**
- On the surface of it, the paper appears to significantly improve query-efficiency over a strong random sampling baseline e.g., 24.44 → 68.2% CIFAR accuracy.
- The core idea is also insightful: leveraging a combination of entropy and diversity within query data.

**3. Related works discussion**
- I particularly liked the extent of the related work section, which faithfully covers most model extraction attacks (including many earlier works).

### Concerns

**1. (Major concern) Selecting hyperparameters / Implementation details**
- A key issue I have with the paper is somewhat insuffient technical details to carefully judge effectiveness of the approach.
- Specifically, I'm unclear on:
    1. what are the hyperparameters for the sampling strategy (e.g., $\gamma_1$, $\gamma_2$, $B$, $k$). Moreover, does this change per dataset?
    2. What is the architecture of the extracted model $f^E$? Is it the same as $f^T$? Does the architecture choice influence results? After all, some signals for the sampling approach is dependent on the architecture (e.g., gradients, losses)
    3. Is the model $f^E$ retrained at each time-step $t$? Is it retrained from scratch? For how many epochs is it trained?
- Importantly, I curious about (1) - how much do choices of the hyperparameters influence results? After all, an reasonable attacker does not have a set to carefully tune it. Can the authors clarify how they searched for the hyperparameters and how much they vary between different datasets?

**2. vs. Defenses**
- Alongside recent attack approaches, there are also a number of recent defenses (e.g., Kariyappa et al., CVPR '20). I'm somewhat surprised that the attack is not evaluated with a defense in place.
- At the very least, I would be curious of the attack performance with a very simple defense strategy e.g., where the target model returns only the argmax prediction label.

**3. Evaluation / Ablation**
- While the experimental results indicate strong query efficiency, there are a few of gaps here:
    1. how influential is each of the proposed sampling strategies (entropy, gradient, loss)? My suggestion is to validate the influence using an ablation study.
    2. a key distinguishing factor between the proposed formulation with prior work is the variational treatment and specifically, the entropy term in Eq. (7). Are the authors aware whether this term leads to empirical gains? Intuitively, I posit that it would encourage sampling around decision boundaries. But since it is estimated on an initially noisy randomly-initialized model, I am not sure how good a signal this results in.
    3. In Fig. 2, why is random sampling a straight line? It appears straight-forward to variably change the size of the random queries and obtain a curve, right?
    4. Table 1 is somewhat unclear: what do rows with a "-", but 100% queries mean?

**4. Disconnect: Formalisms and Approach**
- While I appreciate the formalism on "distributional equivalence", the first part of the paper (Sec. 1-4) led me to different expectations than what's presented later. Specifically, there are multiple claims in the paper on e.g., "generalizing these three approaches using a novel definition ...". However, it appears that the approach and evaluation is tailored on a specific approach on maximizing task accuracy, and somewhat overlooks fidelity, functional equivalence, etc. This was a bit misleading in my opinion.

**4. Misc.**
- Please improve the quality of figures in the paper. Particularly by increasing the font size (highly illegible on paper).
- The authors can complement their related work section by additionally including:
    - data-free model stealing attacks e.g., Truong, Jean-Baptiste, et al. "Data-free model extraction." CVPR '21 (see follow-up works on this as well)
    - Dataset distillation e.g., Wang, Tongzhou, et al. "Dataset distillation." arXiv '18 (who tackle a similar bi-level optimization problem)

**Summary Of The Paper:**

- The paper addresses the problem of model extraction attacks (i.e., learning a surrogate model to mimic a black-box classifier under a budgeted query constraint)
- The main contributions of the paper is two-fold:
    1. formally posing model extraction as a bi-level optimization problem, thereby unifying some work around slightly different extraction notions (fidelity, functional equivalence, ..)
    2. proposing a novel query-efficient approach to solve the above optimization problem. The key idea is to define a set of heuristics to selectively sample data from a query set.
- The approach is evaluated on logistic regression, ResNet-18 (CIFAR dataset), BERT (BBC News) and compared a random sampling strategy. Results indicate significant improvements over random sampling strategy in model extraction and membership inference.

**Summary Of The Review:**

I find the paper studies an interesting problem and also takes a step in the right direction (formal framework to unify recent works, novel query-efficient approach). The key issue I have is that implementation details are unclear (esp. choice of hyperparams) which leads to a concern that query-efficient claims hold only when the attacker is given an oracle test set to tune the attack hyperparameters. I also have a number of different concerns (e.g., attack evaluated against a defense, ablation studies), but I'm somewhat willing to down-weigh these.

---

### Decision · Program_Chairs · 2023-01-20

**Decision:**

Reject

**Justification For Why Not Higher Score:**

See the weakness.

**Justification For Why Not Lower Score:**

N/A

**Metareview: Summary, Strengths And Weaknesses:**

The paper studies blackbox model extraction attacks and focuses on reducing the number of queries to replicate the victim models. The contributions include posing the problem as an online variational optimization problem and proposing a new online and adaptive algorithm based on active learning.

Strength:
+ The paper studies an essential and timely research problem.
+ The reviews appreciate the contribution of the unified framework proposed in the paper. In general, the authors position the paper well in the literature and provide extensive discussion.

Weakness:
- Although the single variation formulation is new, the novelty of the design of Marich is relatively incremental.
- There is a lack of new insight on the proposed approach. A more in-depth discussion about the motivation and analysis are needed to justify the design choices.

Reviewers have mixed judgments about the writing of the paper. The authors have added more experiment details to address the reproducibility concern.




**Summary Of Ac-Reviewer Meeting:**

N/A